# On Sensitivity and Robustness of Normalization Schemes to Input Distribution Shifts in Automatic MR Image Diagnosis

**Divyam Madaan**[1]                  DIVYAM.MADAAN@NYU.EDU
**Daniel Sodickson**[2]           DANIEL.SODICKSON@NYULANGONE.ORG
**Kyunghyun Cho**[1,3,4,5]              KYUNGHYUN.CHO@NYU.EDU
**Sumit Chopra**[1,2]                  SUMIT.CHOPRA@NYU.EDU

[1] *Courant Institute of Mathematical Sciences, New York University, New York, NY*

[2] *Department of Radiology, New York University Grossman School of Medicine, New York, NY*

[3] *Center for Data Science, New York University, New York, NY*

[4] *Prescient Design, Genentech, USA*

[5] *CIFAR Fellow of Learning in Machines & Brains*

**Editors:** Accepted for publication at MIDL 2023

## Abstract

Magnetic Resonance Imaging (MRI) is considered the gold standard of medical imaging because of the excellent soft-tissue contrast exhibited in the images reconstructed by the MRI pipeline, which in-turn enables the human radiologist to discern many pathologies easily. More recently, Deep Learning (DL) models have also achieved state-of-the-art performance in diagnosing multiple diseases using these reconstructed images as input. However, the image reconstruction process within the MRI pipeline, which requires the use of complex hardware and adjustment of a large number of scanner parameters, is highly susceptible to noise of various forms, resulting in arbitrary artifacts within the images. Furthermore, the noise distribution is not stationary and varies within a machine, across machines, and patients, leading to varying artifacts within the images. Unfortunately, DL models are quite sensitive to these varying artifacts as it leads to changes in the input data distribution between the training and testing phases. The lack of robustness of these models against varying artifacts impedes their use in medical applications where safety is critical. In this work, we focus on improving the generalization performance of these models in the presence of multiple varying artifacts that manifest due to the complexity of the MR data acquisition. In our experiments, we observe that Batch Normalization (BN), a widely used technique during the training of DL models for medical image analysis, is a significant cause of performance degradation in these changing environments. As a solution, we propose to use other normalization techniques, such as Group Normalization (GN) and Layer Normalization (LN), to inject robustness into model performance against varying image artifacts. Through a systematic set of experiments, we show that GN and LN provide better accuracy for various MR artifacts and distribution shifts.

**Keywords:** Deep learning, Distribution shifts, MR image diagnosis

## 1. Introduction

Magnetic Resonance Imaging (MRI) (Lauterbur, 1973) is a non-invasive medical imaging technique considered the gold standard of diagnostic imaging because of its excellent soft-tissue contrast and non-ionizing nature. It is an indirect imaging process in which the MR scanner (a complex piece of hardware) collects measurements of the electromagnetic activity within the human subject's body after exposing the subject to a combination of magnetic field and radio frequency pulses. These complex-valued measurements are collected in the frequency space called the $k$-space. The final

grayscale volumetric image is generated from these measurements in two steps. First, a multi-dimensional inverse Fourier transform is used to generate an image in the complex space. Then the grayscale human interpretable image is generated by taking the magnitude of this complex image, which the radiologist reads to render a diagnosis.

While MRI is a vital diagnostic tool, the acquisition process can be challenging. It requires a great deal of expertise to operate the complex hardware, and the process can be time-consuming and costly. A wide range of acquisition parameters must be carefully controlled and optimized to obtain high-quality images, including tissue and scanner settings, sequence types, and parameters. This complexity can lead to various image artifacts (Sled and Pike, 1998; Heiland, 2008), both within a single machine and across different machines and patients. Another significant challenge of MRI is the sensitivity to motion artifacts (Zaitsev et al., 2015; Shaw et al., 2019; Lee et al., 2020; Wang et al., 2020). Due to the long acquisition time, even small movements by the patient can result in ghosting, blurring, and affine transformations in the images. Additionally, when training systems for automatic medical diagnosis, these artifacts can cause discrepancies in the training and validation datasets, as they are often removed from the collected data, making them out of distribution.

More recently, Deep neural networks (DNNs) have shown great potential in improving various aspects of the MRI pipeline, including 1) speeding up the measurement acquisition process by generating diagnostic quality images using under-sampled $k$-space data (Vellagoundar and Machireddy, 2015; Schlemper et al., 2018; Gözcü et al., 2019; Bakker et al., 2020), and 2) performing pathology segmentation or inferring the presence/absence of diseases from the volumetric grayscale images to assist radiologists in more accurate and faster diagnosis (Hammernik et al., 2018; Knoll et al., 2019; Liang et al., 2019; Putzky and Welling, 2019; Sriram et al., 2020). Despite the impressive performance of DNNs in disease identification on retrospective data sets, their use in real clinical practice has remained elusive. Among the many reasons contributing to this discrepancy, one of them is the sensitivity of these models to changes in input data distribution shifts (Goodfellow et al., 2014; Hendrycks and Dietterich, 2019; Geirhos et al., 2019). Unfortunately, as discussed in the previous paragraph, such input distribution shifts are quite prevalent in the MR imaging pipeline due to multiple noise sources in the acquired measurements. This leads to artifacts within the reconstructed images, varying across patients, machines, and even within the same machine.

With an over-arching goal of making DNNs suitable for use within real clinical practice, in this research, we try to better understand the source(s) behind such brittleness of DNN models. Upon extensive literature review, we identify at least one common underlying theme that connects all the works involving DNNs for medical imaging tasks. We observed that across the board, these models are trained using Batch Normalization (BN) (Ma and Jia, 2020; Sharma et al., 2021; Chiang et al., 2021; Wahlang et al., 2022; Mallya and Hamarneh, 2022; Chikontwe et al., 2022), normalization methodology extensively used in successfully training DNNs for a variety of tasks involving natural images (Szegedy et al., 2016; He et al., 2016a; Hu et al., 2018; Sandler et al., 2018). We posit that BN contributes to model vulnerability when dealing with distribution shifts during automatic diagnosis. Specifically, BN statistics obtained from the training phase are sub-optimal during inference when evaluated with various out-of-distribution scenarios, such as MR artifacts resulting from the data acquisition process. To support this argument, we focus on the task of identifying multiple clinically significant pathologies from the MR scans and simulate various artifacts (see Figure 1), including herringbone artifact, Rician noise artifact (Rice, 1944; Gudbjartsson and Patz, 1995), biased field intensity (Sled and Pike, 1998), and subject-related motion artifacts (Zaitsev et al., 2015; Godenschweger et al., 2016; Shaw et al., 2019; Lee et al., 2020) on the publicly

available fastMRI dataset (Zbontar et al., 2018; Zhao et al., 2021). Our experiments reveal that batch normalization leads to a significant degradation in the model performance with up to $10\%$ drop on the AUROC (BRA, 1997) for distribution shifts caused by certain artifacts.

These results are particularly noteworthy because, until now, the research community in this field has focused on training DNNs primarily using batch normalization for automated diagnosis. Recently, techniques such as Group Normalization (GN) (Wu and He, 2018) and Layer Normalization (LN) (Ba et al., 2016) have gained popularity for image segmentation (Kao et al., 2019; Zhou and Yang, 2019; Chen et al., 2021). However, to the best of our knowledge, this is the first work to demonstrate their robustness to various changes and artifacts in the MR disease identification task. To better understand these observations, we compare the BN statistics computed during training with the statistics of the test samples and observe that the training statistics do not align with the test environment. Additionally, we found that adapting the statistics during testing can improve performance; however, alternate normalization strategies are crucial for ensuring the robustness and applicability of these models in real-world clinical scenarios with changing conditions. Our findings provide new insight for the community to consider different normalization strategies for DNNs in medical imaging. The key contributions of our work are:

- We highlight the susceptibility of batch normalization in the task of disease prediction when encountering distribution shifts and various artifacts present in practical clinical scenarios of Magnetic Resonance Imaging (MRI).

- We show that alternate normalization strategies, such as group normalization and layer normalization for intermediate layers, are more robust compared to batch normalization and essential for training deep neural networks that are more robust to these issues.

- We further explore the reasons for the susceptibility of batch normalization.

## 2. Related Work

**Application of deep learning to MRI.**    Over the last years, DNNs have experienced significant success in solving image reconstruction and image analysis problems within the MRI pipeline. Image reconstruction involves improving the efficiency of this process by reconstructing diagnostic quality images from a fraction of the frequency space data, thereby significantly reducing the burden on the data acquisition process (Lustig et al., 2007; Vellagoundar and Machireddy, 2015; Schlemper et al., 2018; Gözcü et al., 2018; Haldar and Kim, 2019; Gözcü et al., 2019; Sanchez et al., 2020; Bakker et al., 2020). For image analysis, DNNs have been trained to perform tasks like tissue/organ segmentation or disease identification with high accuracy (Schmah et al., 2008; Liu et al., 2014; Hatakeyama et al., 2014; Moeskops et al., 2016; Milletari et al., 2016; Chen et al., 2016; Hwang and Kim, 2016; Pinaya et al., 2016; Lakhani and Sundaram, 2017; Park et al., 2019; Kim et al., 2019). While successful, the evaluation of these models has been limited to carefully collected retrospective data sets where special attention is paid to ensure that the input samples in the test set have the same distribution as the input distribution of the training set.

**Distribution shifts in MR image diagnosis.**    Previous research in computer vision (Goodfellow et al., 2014; Hendrycks and Dietterich, 2019; Geirhos et al., 2019) has revealed that DL classification models are vulnerable to distribution shifts, which can negatively impact their performance. On the other hand, the medical imaging community has mainly focused on the robustness of segmentation

models (Karani et al., 2021; Zhang et al., 2020; Yan et al., 2020; Tomar et al., 2022) to understand the impact of anatomy, dataset, modality, and acceleration shifts. However, collecting data from different scanners and medical sites is a difficult task due to the sensitive nature of patient data. Additionally, there is still much to be learned about the robustness of deep-learning models for disease identification. Therefore, in this work, we investigate the impact of input distribution shifts (Sled and Pike, 1998; Zaitsev et al., 2015; Godenschweger et al., 2016) that might occur in actual clinical settings because of the complexities associated with the MR data acquisition process on the performance of DNNs.

## 3. Automatic MR Image Diagnosis with Input Distribution Shifts

### 3.1. Motivation

The MR data acquisition process is multi-faceted, with a long-acquisition time that combines individual signals of different contrasts and adjusts various parameters such as proton density, temperature, and field strength. However, the multi-parameter dependency can result in unwanted noise and artifacts in the acquired MR signals, which can make interpretation of the images challenging, especially in cases where diagnostic accuracy is critical.

The long acquisition times also lead to many clinical and research application challenges. For example, patients may find the process uncomfortable or claustrophobic, and the procedure may not be suitable for patients with certain medical conditions. Efforts have been made to reduce the acquisition time (Lustig et al., 2007; Vellagoundar and Machireddy, 2015; Schlemper et al., 2018; Sanchez et al., 2020; Bakker et al., 2020), but these methods reduce the image resolution, contrast, and signal-to-noise ratio. These challenges are further exacerbated by the use of low-field scanners, which despite being inexpensive, have yet to see widespread adoption due to the poor diagnostic quality of the images generated by them.

Therefore, to ensure the effectiveness of deep learning models in clinical applications, evaluating them in real-world scenarios is crucial. This will provide a better understanding of the model's ability to generalize and perform in actual situations. In addition, such evaluations can aid researchers and practitioners in identifying the limitations of the models and how they can be improved to serve patients and healthcare providers better.

### 3.2. Problem Setup

We consider the task of disease prediction, where we are given the ground-truth MR images denoted by $\mathcal{D} = \{(x_i, y_i)_{i=1}^{\mathcal{N}}\}$ with $\mathcal{N}$ examples and $K$ pathologies. We consider a classifier $f_\Theta : \mathbb{R}^d \to K$ with $L$ layers and $\Theta$ as parameters, optimize the cross-entropy loss $\mathcal{L}$ by minimizing the expected loss over the training data distribution $\mathcal{D}_{tr}$, mathematically represented as:

$$\underset{\Theta}{\text{minimize}} \; \mathbb{E}_{(x,y) \sim \mathcal{D}_{tr}} \Big[ \mathcal{L}(f_\Theta(x), y) \Big]. \tag{1}$$

However, input data distribution can change during inference for various reasons in clinical applications. In this work, we focus on model evaluation in the presence of covariate shifts (Sugiyama and Kawanabe, 2012; Schölkopf et al., 2012), where the conditional distribution $p_{\text{train}}(y|x) = p_{\text{test}}(y|x)$, but $p_{\text{train}}(x) \neq p_{\text{test}}(x)$. We denote the clean and corrupted images with $x$ and $\widehat{x}$, respectively.

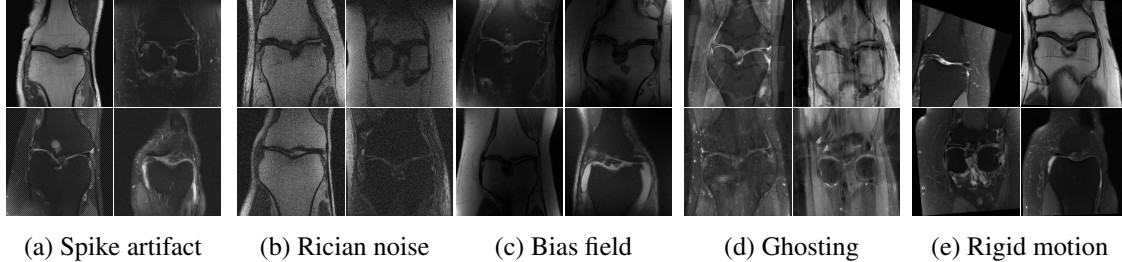

| (a) Spike artifact | (b) Rician noise | (c) Bias field | (d) Ghosting | (e) Rigid motion |

Figure 1: Visualization of simulated artifacts for the ground truth images.

### 3.3. Simulation of Various Artifacts

Obtaining corrupted MR images through traditional data acquisition methods can take time and resources. To overcome this limitation, we simulate various MR artifacts to evaluate the generalization of DNNs in the presence of input distribution shifts while minimizing the burden on data acquisition. Furthermore, by simulating different types of MR artifacts, we can create a diverse dataset to evaluate DNNs, which can help improve their performance on real-world MR images.

**Hardware-related artifacts.** Technical defects in the MR acquisition components, such as gradient coils and amplifiers, cause these artifacts. One typical example of such artifacts is the *spike or herringbone artifact*, created by electronic discharge in the receiver chain. We manifest this artifact as wave-like fringes within the MR image (see Figure 1a) and change the ratio between the spike intensity and the spectrum maximum to vary the intensity levels.

**Rician noise artifact.** Rician noise artifact arises due to the change in the field strength of MRI scanners. It is caused by the Gaussian distributed noise present in raw *k*-space data, which leads to ground-truth images with noise that follows a Rician distribution (Rice, 1944; Gudbjartsson and Patz, 1995) as shown in Figure 1b. Mathematically, for SNR $k$, this can be formalized as $\widehat{x} = |x + \mathcal{N}_1(0, \sigma) + i\mathcal{N}_2(0, \sigma)|$, where $\sigma = $ signal strength$/k$, signal strength is the maximum pixel intensity for each instance.

**Intensity non-uniformity.** This artifact is caused by variations in intensity due to inhomogeneous RF excitation field, non-uniform reception coil sensitivity, RF penetration, and standing-wave effects (Sled and Pike, 1998). The resulting bias field can be modeled as a linear combination of polynomial basis functions (Van Leemput et al., 1999) as shown in Figure 1c. In this work, we model the bias field as a third-order multiplicative polynomial and vary the maximum magnitude of the coefficients to change the intensity of the field.

**Subject-related artifacts.** Due to the long acquisition time, MRI is sensitive to the movement of the subject that causes motion artifacts (Zaitsev et al., 2015; Godenschweger et al., 2016; Shaw et al., 2019; Lee et al., 2020). *Rigid motion* is caused due to the random movement of the subject. It can be observed in all body parts and quantified by translation and rotation parameters (see Figure 1e), which are commonly used in computer vision when working with natural images. We vary the translation between 0 and 10 mm and angle between 0 to $20°$ (Wang et al., 2020).

*Non-rigid motion* is caused by involuntary motion, cardiac and respiratory motion, vessel pulsation, gastrointestinal peristalsis, and blood and CSF flow. This produces ghosting artifacts – partial or complete replication of the subject along the phase-encoding dimensions, blurring, or decreased SNR ratio. We simulate the ghosting effect using $N$ distinct ghosts by zeroing every $N^{th}$ plane in k-space in the phase-encoding axis as shown in Figure 1d. The ghosting axis was randomly chosen and intensity $s$ was varied between $(0, d)$, such that $s \sim \mathcal{U}(0, d)$ for all pathologies.

### 3.4. Feature Normalization

Over the past years, feature normalization has emerged as an important component for the faster convergence and stability of training deep learning models. The general formulation of feature normalization for features $a_i^\ell \in \mathbb{R}^{N \times C \times H \times W}$ involves computing the mean $\mu_i$ and standard deviation $\sigma_i$ of the features and then using these statistics to normalize the features. Specifically, the normalized version of a feature $a_i^\ell$ can be defined following Wu and He (2018):

$$\widehat{a}_i^\ell = \frac{1}{\sigma_i}(a_i - \mu_i), \ \ \text{where} \ \ \mu_i = \frac{1}{m}\sum_{k \in \mathcal{S}_i} a_k, \ \ \sigma_i = \sqrt{\frac{1}{m}\sum_{k \in \mathcal{S}_i}(a_k - \mu_k)^2 + \epsilon} \qquad (2)$$

**Batch normalization (BN)** (Ioffe and Szegedy, 2015) is a powerful technique widely used in deep learning models to improve their performance. The method normalizes the intermediate layer's features by utilizing the statistics computed across the training data, $\mathcal{S}_i = \{k | k_C = i_C\}$. However, when the distribution of the test data changes, these pre-computed statistics may not be optimal. Additionally, using smaller batch sizes during training can result in less accurate estimates.

**Adaptive Batch normalization (AdaBN)** (Li et al., 2016) is an extension of the standard Batch Normalization technique, which recomputes the BN statistics using an exponential moving average across the test distribution to improve model generalization. While prior works (Schneider et al., 2020; Nado et al., 2020; Benz et al., 2021) have shown that it addresses the issue of input distribution shifts, it is not sufficient to obtain robust models for clinical diagnosis. In particular, when the distribution shift is severe, AdaBN may even degrade the model's performance (see Figure 2).

**Layer normalization (LN)** (Ba et al., 2016) is an alternative normalization technique to Batch Normalization and AdaBN. LN calculates the mean and standard deviation across all features in a layer instead of only using the batch dimension as in Batch Normalization, $\mathcal{S}_i = \{k | k_N = i_N\}$. This allows LN to be applied consistently in both the training and testing phases, making it less sensitive to the data distribution. Additionally, Layer normalization is independent of re-scaling and shifting of individual training examples.

**Group normalization (GN)** (Wu and He, 2018) divides channels into $G$ groups and computes the mean and standard deviation across groups. $\mathcal{S}_i = \{k | k_N = i_N, \lfloor \frac{k_C}{C/G} \rfloor = \lfloor \frac{i_C}{C/G} \rfloor\}$. The independence of GN from the batch dimension makes it less sensitive to the distribution of the data, which allows for better generalization performance. The multiple groups also provide a way for the model to learn different distributions for each group, which increases flexibility compared to Layer Normalization, where the number of groups is set to one.

## 4. Experiments

### 4.1. Experimental Setup

**Data Sets.** We consider the *knee pathologies* from the *fastMRI* (Zbontar et al., 2018) dataset, where we use the ground-truth images with slice-level labels using fastMRI+ (Zhao et al., 2021) for the three most significant pathologies as suggested by the clinicians. Concretely, we consider 1) *Anterior Cruciate ligament (ACL)* with 1,443 annotations of 254 subjects, 2) *Meniscus tear* with 5,658 annotations of 663 subjects, and 3) *Cartilage* with 3,600 annotations of 710 subjects. The slices were cropped to $320 \times 320$ and $15\%$ of the dataset for validation and test sets.

**Models and Metrics.** We conduct our experiments with PreactResNet-18 (He et al., 2016a) using different normalization schemes as described in §3.4. We report the mean and standard deviation across five independent runs. We evaluate with two commonly-used and widely accepted metrics, AUROC (BRA, 1997) and Balanced Accuracy (García et al., 2009) on the varying intensity of artifacts described in §3.3. The details for these metrics, code, hyper-parameters and the description for the intensity levels of various artifacts are provided in Appendix A.

### 4.2. Quantitative Results

**Evaluation on various artifacts.** Figure 2 shows the evaluation of the model trained with clean images on various artifacts. The results indicate that AdaBN is a simple yet effective method that improves the AUROC on the spike and Ricean noise artifacts compared to traditional BN. However, it is less effective on other types of artifacts. In contrast, GN and LN techniques overcome the limitations of AdaBN and BN by demonstrating superior performance across all artifacts. The most pronounced improvement is observed for spike and Ricean noise, resulting in an absolute increase of $10 - 15\%$ in AUROC at higher noise intensities. Similar performance gains are also observed for field and ghosting artifacts, with an absolute improvement of $8\%$ and $4\%$, respectively. Interestingly, BN performs similarly to GN and LN when dealing with rigid-motion artifact. Further, we report the balanced accuracy metric in Appendix B. Furthermore, in Figure C.13 and Figure C.12, we conduct a comparison between models that were trained using Instance Normalization (IN) (Ulyanov et al., 2016) and those that underwent no normalization. It is worth noting that for certain artifacts, the model with no normalization achieved better performance than BN, while GN and LN outperformed IN for most artifacts.

It is worth noting that AdaBN requires access to a set of test examples for adaptation, while GN or LN only requires a single test example during inference. This makes GN and LN more practical for real-world scenarios where the availability of examples from out-of-distribution shifts may be limited. Overall, our evaluation suggests that GN and LN are more robust normalization schemes for improving the performance of a model trained on clean images when dealing with various artifacts.

**Further analysis.** To understand the behavior of batch normalization (BN), we compare the statistics calculated during training with the input statistics during testing. Specifically, for the mean $\mu_i$ and variance $\sigma_i^2$ of layer $i$, we calculate the $\ell_2$ distance between the mean and variance of the test data and BN statistics ($\|\mu_h - \mathbb{E}_{test}[h]\|^2$, $\|\sigma_h^2 - \text{Var}[h]\|^2$) in Figure 3. The results show that the spike and Ricean artifacts have a more significant impact on the difference in the variance of the BN layer from the test distributions compared to other shifts. This is consistent with our evaluations in Fig-

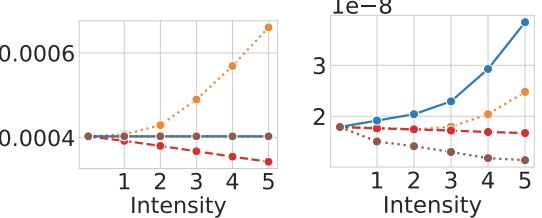

Figure 3: $\ell_2$ distance between first batch norm layer statistics and inference data statistics for various shifts (mean on the left and variance on right) for **Ricean artifact**, **motion artifact**, **spike artifact** and **ghosting artifact**. We observe that performance degradation for BN models is consistent with the increase in the drift of BN statistics from the test distributions.

ure 2, which have shown a significant decline in the performance of models trained with BN under these distributions. We observed that the Ricean artifact has a more substantial change in the mean compared to other artifacts. We defer the partial adaptation of BN statistics to Appendix B. Further, Figure C.14 shows a consistent degradation with BN across all shifts irrespective of the batch size.

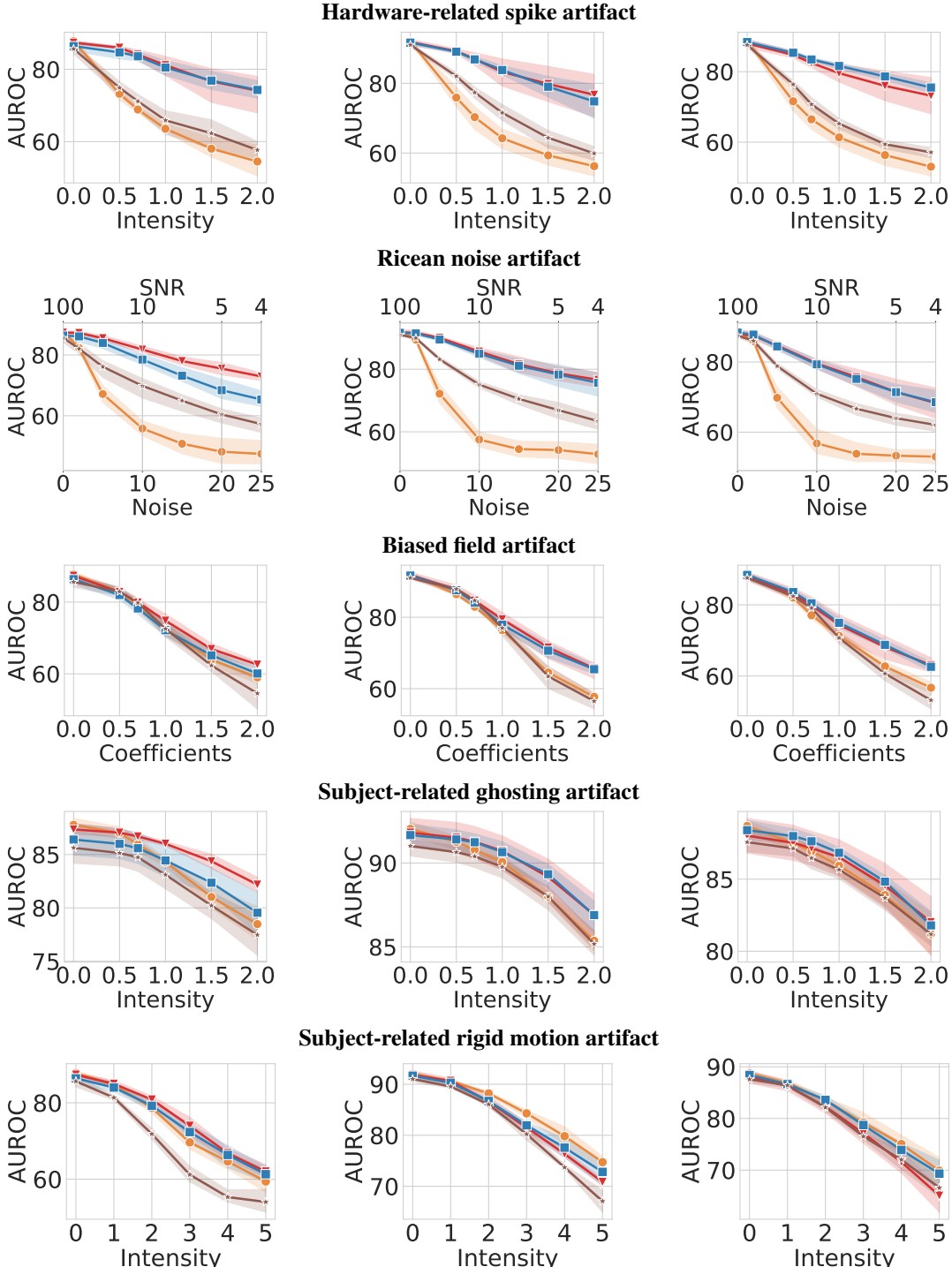

Figure 2: Comparison of PreactResNet-18 trained with **batch normalization**, **group normalization**, **layer normalization**, and **adaptive batch normalization** on ACL (**left**), Meniscus Tear (**middle**), and cartilage (**right**). We observe that GN and LN obtain better or comparable performance for majority of the shifts.

## 5. Conclusion

In this research, we investigate the robustness of DNNs when applied to disease identification. To answer our research question, we conduct an empirical analysis to examine their effectiveness against various artifacts commonly found in MRI such as hardware-related spiking artifact, Ricean artifact, biased-field effect, and subject-related motion artifacts on fastMRI knee pathology data. Our findings revealed that Batch Normalization, a widely used technique, is a significant contributing factor to the sensitivity of DNNs to these artifacts. This is because the BN statistics computed during training may not be optimal for handling distribution shifts that occur at test time. To tackle this problem, we explored alternative normalization methods such as Group Normalization and Layer Normalization. We found that these methods are more robust to input distribution shifts. In addition, we compared the BN statistics computed during training with the statistics of the test-time input distribution, and found that the performance drop of the model was proportional to the deviation between the two. Our findings have several implications for developing more robust DNNs for medical imaging and can potentially improve their accuracy and reliability of automatic disease identification.

## Acknowledgement

We thank the anonymous reviewers for their insightful comments and suggestions. This research was supported in part by the Center for Advanced Imaging Innovation and Research (CAI2R), a National Center for Biomedical Imaging and Bioengineering operated by NYU Langone Health and funded by the National Institute of Biomedical Imaging and Bioengineering through award number P41EB017183. This content is solely the responsibility of the authors and does not necessarily represent the official views of the National Institutes of Health.

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

# Supplementary Material
# On Sensitivity and Robustness of Normalization Schemes to Input Distribution Shifts in Automatic MR Image Diagnosis

**Organization.** The appendix is organized as follows: We provide the experimental details in Appendix A. Next, we provide additional results and visualizations in Appendix B and additional related work in Appendix C. The code is publicly available online.

## Appendix A. Experimental Details

We extend the fastMRI (Zbontar et al., 2018) open-source codebase[1] to the task of disease classification for our experiments. We train all the methods with a batch size of 32 with early stopping using mean validation AUC. We do a grid-search of learning rate in $[1e-5, 1e-4, 1e-3, 1e-2]$ and weight decay in $[1e-5, 1e-4, 1e-3, 1e-2, 1e-1]$ for all the methods. We use a single NVIDIA Quadro RTX 8000 for conducting all the normalization schemes. We evaluate all our experiments with:

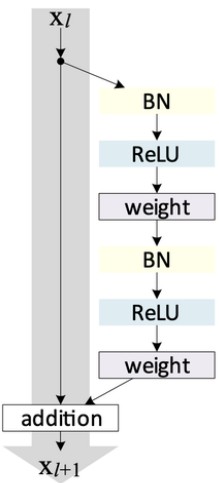

1. **AUROC** is a common metric used for the evaluation of medical problems solved in machine learning (Rajkomar et al., 2018; Rajpurkar et al., 2017; Solares et al., 2020; Tschandl et al., 2019; Teixeira et al., 2017; Dunnmon et al., 2019). It computes the area under the receiver operating characteristic curve and can be calculated by the area under the False Positive Rate (FPR) against the True Positive Rate (TPR) curve.

2. **Balanced Accuracy (García et al., 2009).** Due to the class-imabalance across pathologies in the knee data set, we use this metric as it gives equal weight to both majority and minority classes and is commonly used in the presence of class imbalance learning (Khan et al., 2019). It is computed as the average of the sensitivity and specificity, where sensitivity and sensitivity are the true positive rate and true negative rate respectively.

Figure A.4: PreactResNet block following He et al. (2016b).

**Architecture details.** We adhere to the standard PreactResNet18 architecture (He et al., 2016b) as shown in Figure A.4 and substitute the batch normalization layers with alternate normalization layers.

**Artifact details.** We implement the Ricean artifact and utilize the TorchIO (Pérez-García et al.) library to simulate other artifacts. The following details the parameters for varying intensities across different artifacts:

- **Hardware-related spike artifact** has two key elements – the number of spikes and the intensity. The intensity is defined as the ratio between the spike intensity and the maximum of the spectrum. The number of spikes is selected randomly from a uniform distribution between zero and the maximum number of spikes, and the intensity $d$ is varied between $d \in \{0.5, 0.7, 1.0, 1.5, 2.0\}$ for all the pathologies. We visualize the different intensities in Figure C.5.

---

1. https://github.com/facebookresearch/fastMRI

- **Ricean noise artifact** has the SNR of the images as the hyper-parameter. The noisy images generated by the SNR values considered for this artifact are visualized in Figure C.6.

- **Biased field artifact** has two hyper-parameters – the order of the basis polynomial function and the maximum magnitude of polynomial coefficients. The order is fixed at three, and the maximum coefficient varies between $\{0.5, 0.7, 1.0, 1.5, 2.0\}$ (see Figure C.7).

- **Subject-related ghosting artifact** has two hyper-parameters – the number of ghosts and the artifact strength in relation to the k-space maximum. We fix the number of ghosts to seven and vary the strength between $\{0.5, 0.7, 1.0, 1.5, 2.0\}$. We visualize these variations in Figure C.8.

- **Subject-related rigid motion artifact.** has two parameters - the rotation range of the simulated movements (in degrees) and the translation in mm of the simulated movements. The translation varies between $\{2, 46, 8, 10\}$ and the rotation range varies between $\{5, 10, 15, 20, 25\}$ following Lee et al. (2020). These variations are shown in Figure C.9.

## Appendix B. Additional Experiments

The balanced accuracy metric, a measure of the model's overall performance is shown in Figure C.10 for various artifacts using models trained with different normalization schemes. Similar to the AUROC metric in Figure 2, we observe consistent results for all evaluated artifacts for balanced accuracy. This indicates that the model's performance is stable and consistent across different types of artifacts.

Additionally, Figure C.11 illustrates the results for the partial adaptation of BN statistics. The results demonstrate that adapting both the mean and variance is generally more effective than adapting only one for most of the artifacts considered.

## Appendix C. Additional Related Work

The studies conducted by Summers and Dinneen (2020) and Singh and Shrivastava (2019) aimed to enhance the performance of BN by addressing its limitations, mainly when working with small batch sizes. To achieve this, these studies leveraged various techniques, including inference level statistics and the impact of weight decay on the scaling and shifting parameters of BN. Other research in this field, such as the works by Henaff (2020) and Rivoir et al. (2022) also highlight the limitations of BN in other contexts. For instance, Henaff (2020) showed that BN could negatively impact performance with Contrastive Predictive Coding, while LN is a more practical alternative. Meanwhile, Rivoir et al. (2022) investigated the robustness of BN in end-to-end video learning. Their findings showed that CNN-LSTMs without BN outperformed the current state of the art in surgical phase recognition.

In contrast, our work is different from the previous work in multiple ways. First, our research suggests that BN and Adaptive BN are not always effective in mitigating the artifacts likely to be present in MR images, particularly when the data distribution shift due to the artifacts is large: we show that alternative normalization techniques are more effective in these scenarios. Second, our study exclusively focuses on medical imaging tasks (specifically MR imaging), where the research community has primarily relied on BN for training DNNs (evident from the state-of-the-art references cited in our work, showcasing the widespread use of BNs in the medical field). In that sense our work can be seen as an important extension of the previous works, highlighting the deficiencies of BN in medical imaging tasks.

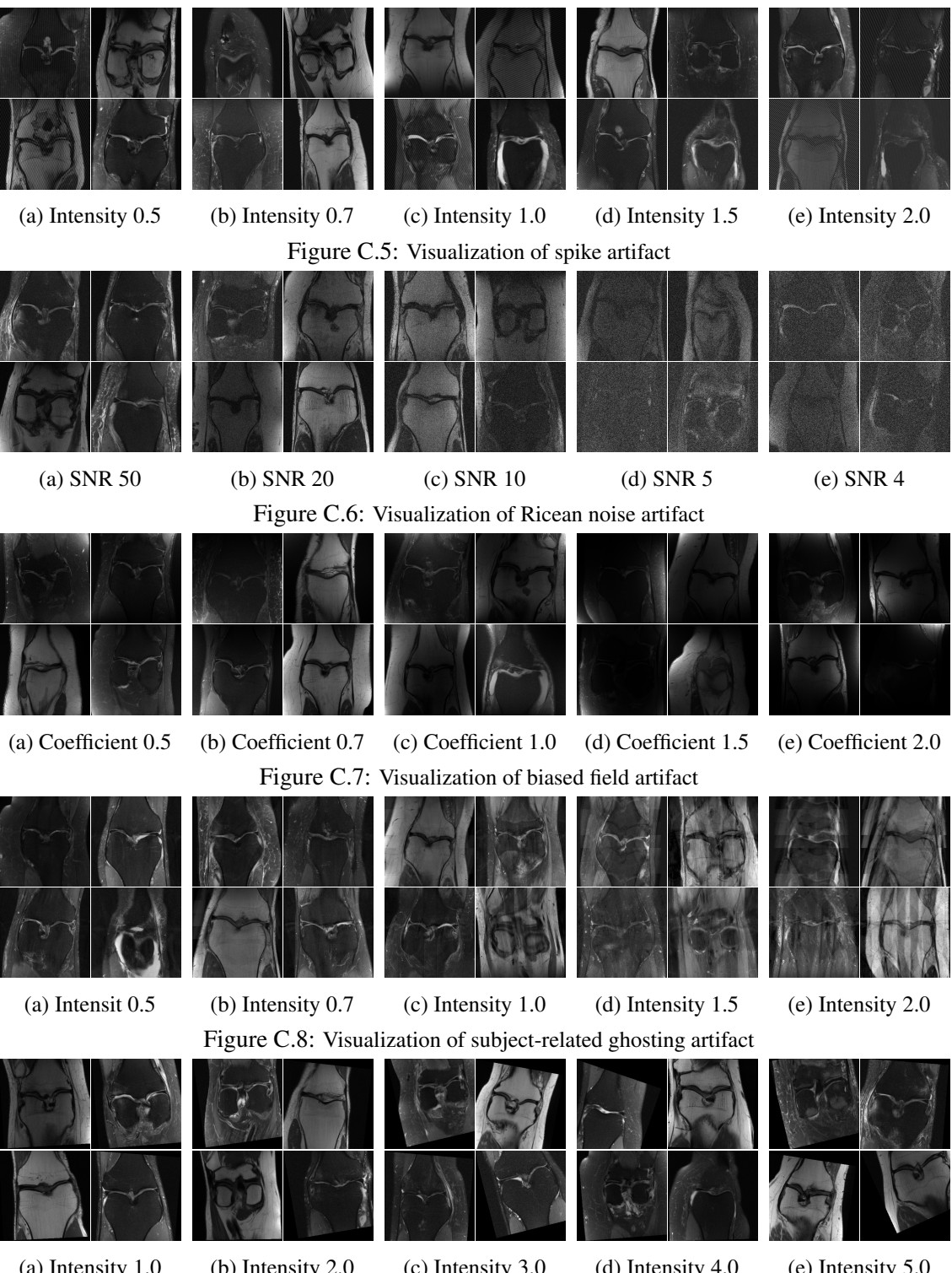

(a) Intensity 0.5  (b) Intensity 0.7  (c) Intensity 1.0  (d) Intensity 1.5  (e) Intensity 2.0

Figure C.5: Visualization of spike artifact

(a) SNR 50  (b) SNR 20  (c) SNR 10  (d) SNR 5  (e) SNR 4

Figure C.6: Visualization of Ricean noise artifact

(a) Coefficient 0.5  (b) Coefficient 0.7  (c) Coefficient 1.0  (d) Coefficient 1.5  (e) Coefficient 2.0

Figure C.7: Visualization of biased field artifact

(a) Intensit 0.5  (b) Intensity 0.7  (c) Intensity 1.0  (d) Intensity 1.5  (e) Intensity 2.0

Figure C.8: Visualization of subject-related ghosting artifact

(a) Intensity 1.0  (b) Intensity 2.0  (c) Intensity 3.0  (d) Intensity 4.0  (e) Intensity 5.0

Figure C.9: Visualization of subject-related rigid motion artifact.

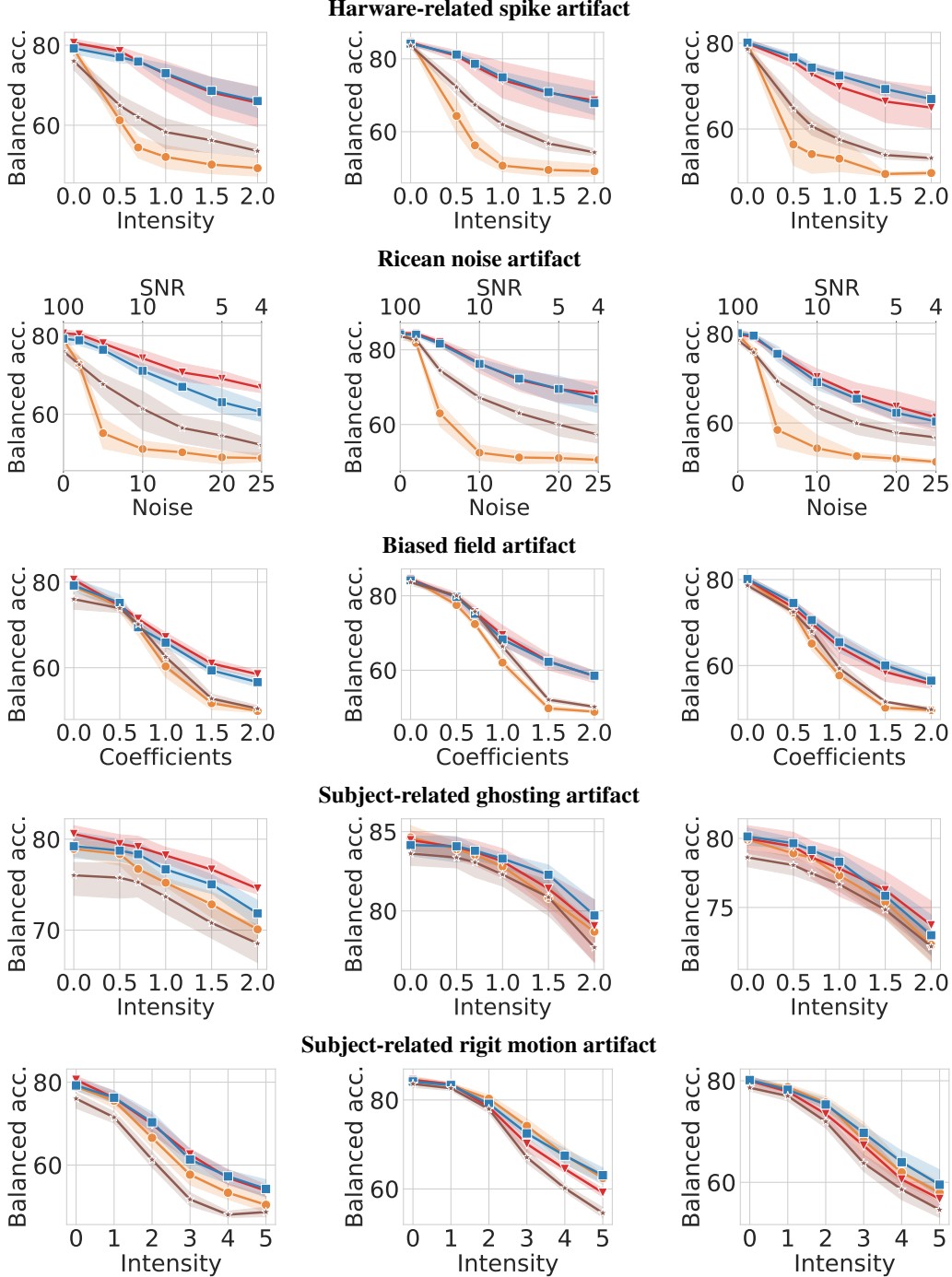

Figure C.10: Balanced accuracy comparison of PreactResNet-18 trained with **batch normalization**, **group normalization**, **layer normalization**, and **adaptive batch normalization** on ACL (**left**), Meniscus Tear (**middle**), and cartilage (**right**).

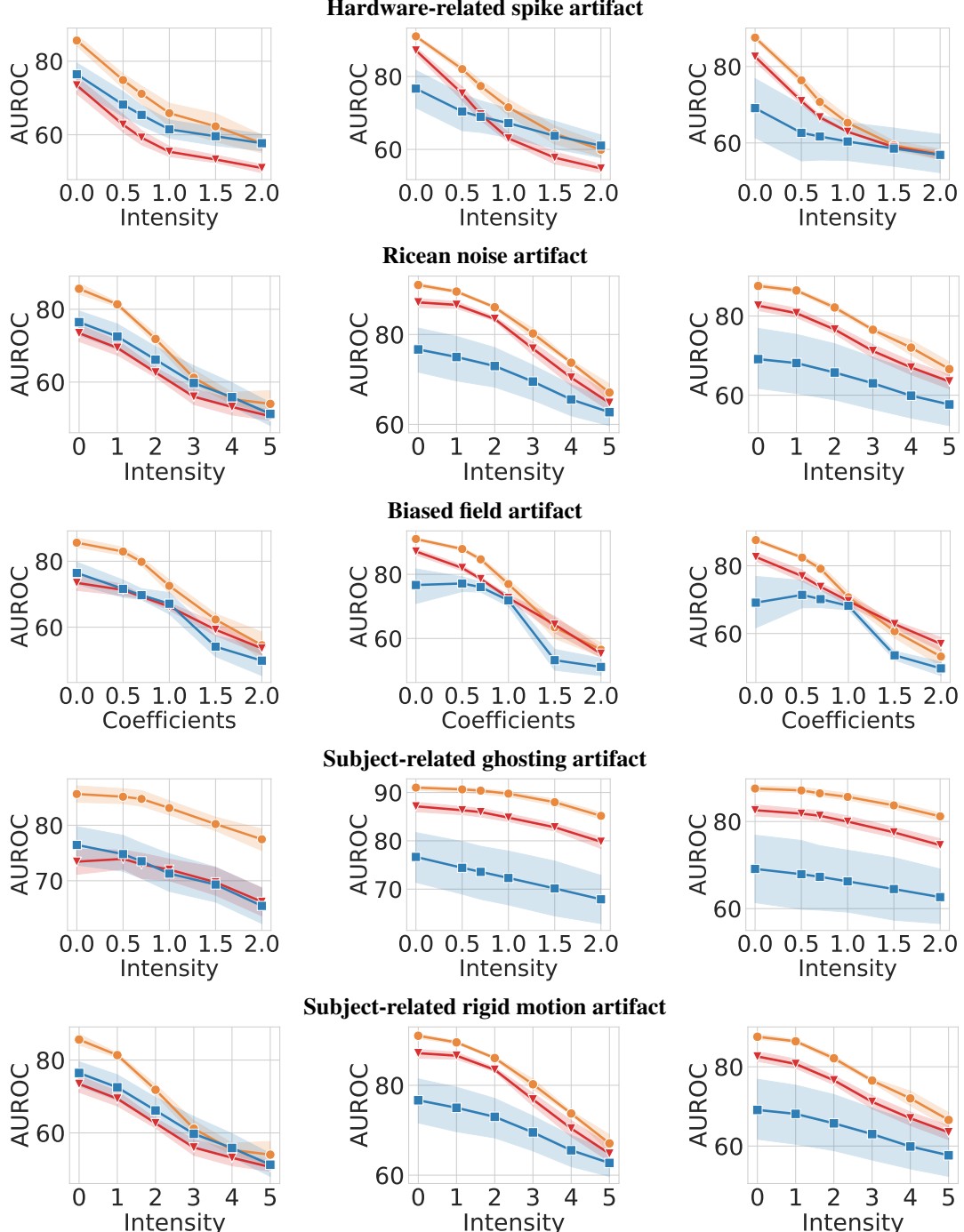

Figure C.11: Comparison of PreactResNet-18 trained adapted with **both mean and variance**, **mean only**, and **variance only** on ACL (**left**), Meniscus Tear (**middle**), and cartilage (**right**) for grounth truth images. We observe that adapting both is beneficial for majority of the artifacts.

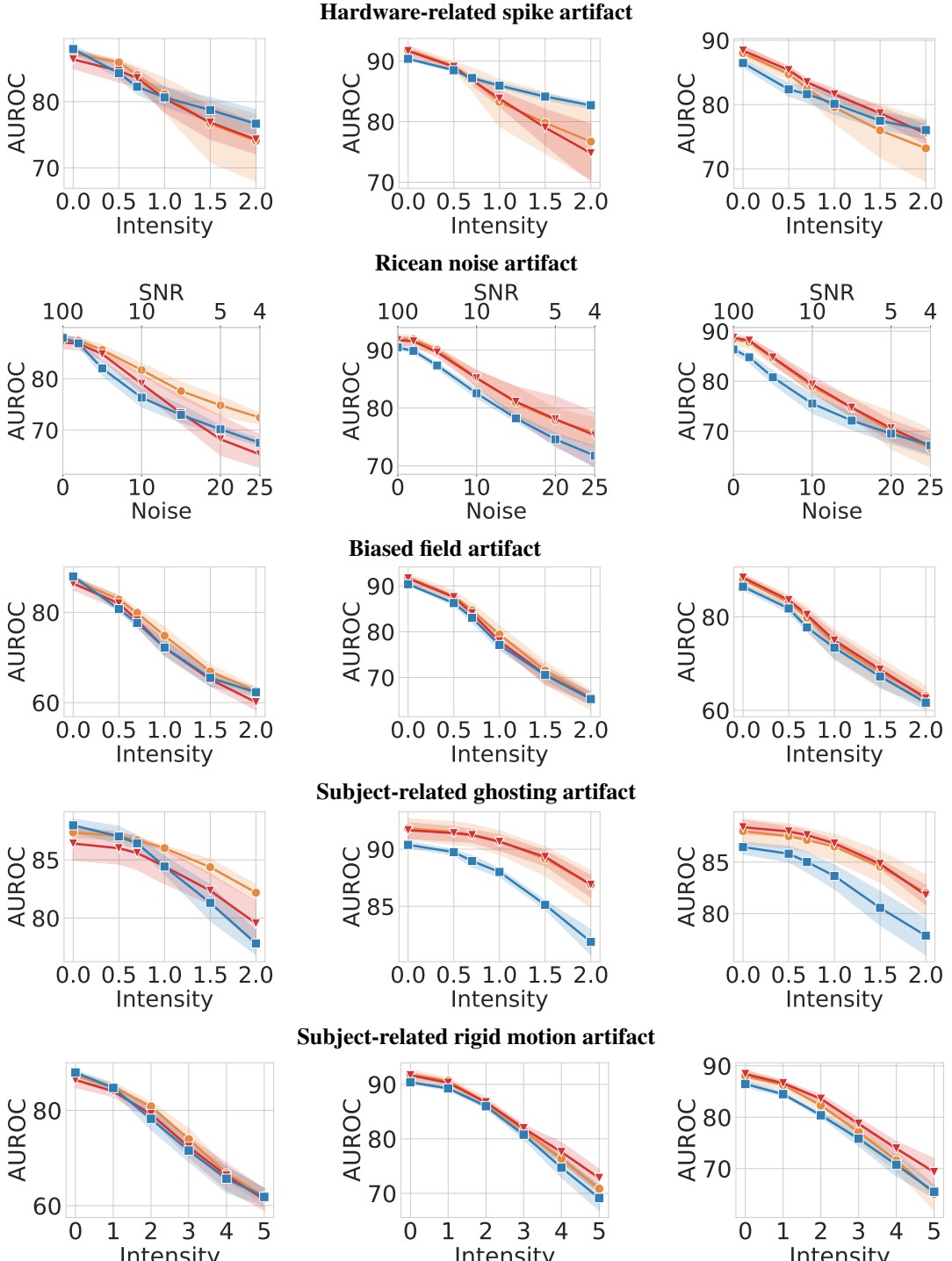

Figure C.12: Comparison of PreactResNet-18 trained with **group normalization**, **layer normalization**, and **instance normalization** on ACL (**left**), Meniscus Tear (**middle**), and cartilage (**right**).

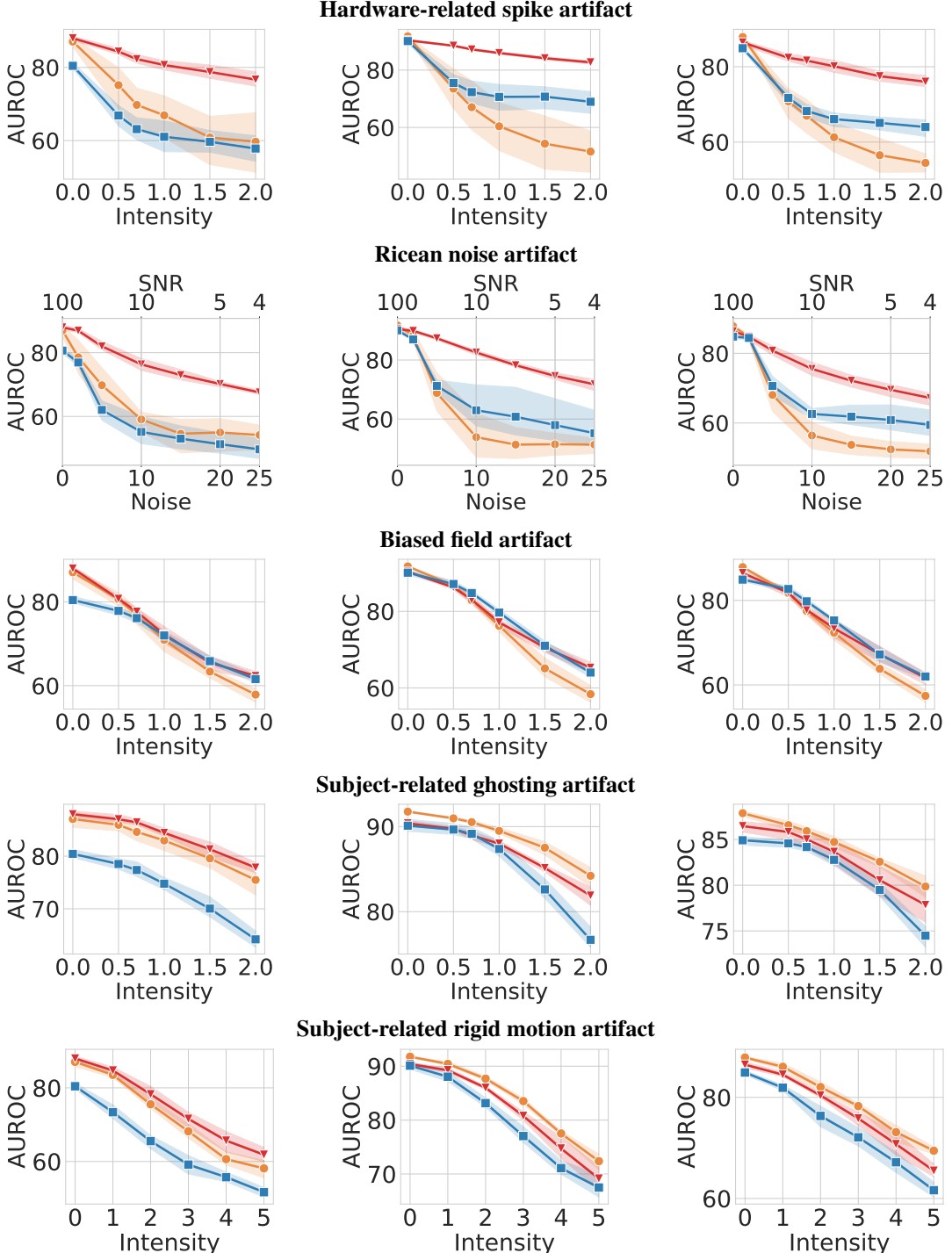

Figure C.13: Comparison of PreactResNet-18 trained with **batch normalization**, **instance normalization**, and **no normalization** on ACL (**left**), Meniscus Tear (**middle**), and cartilage (**right**).

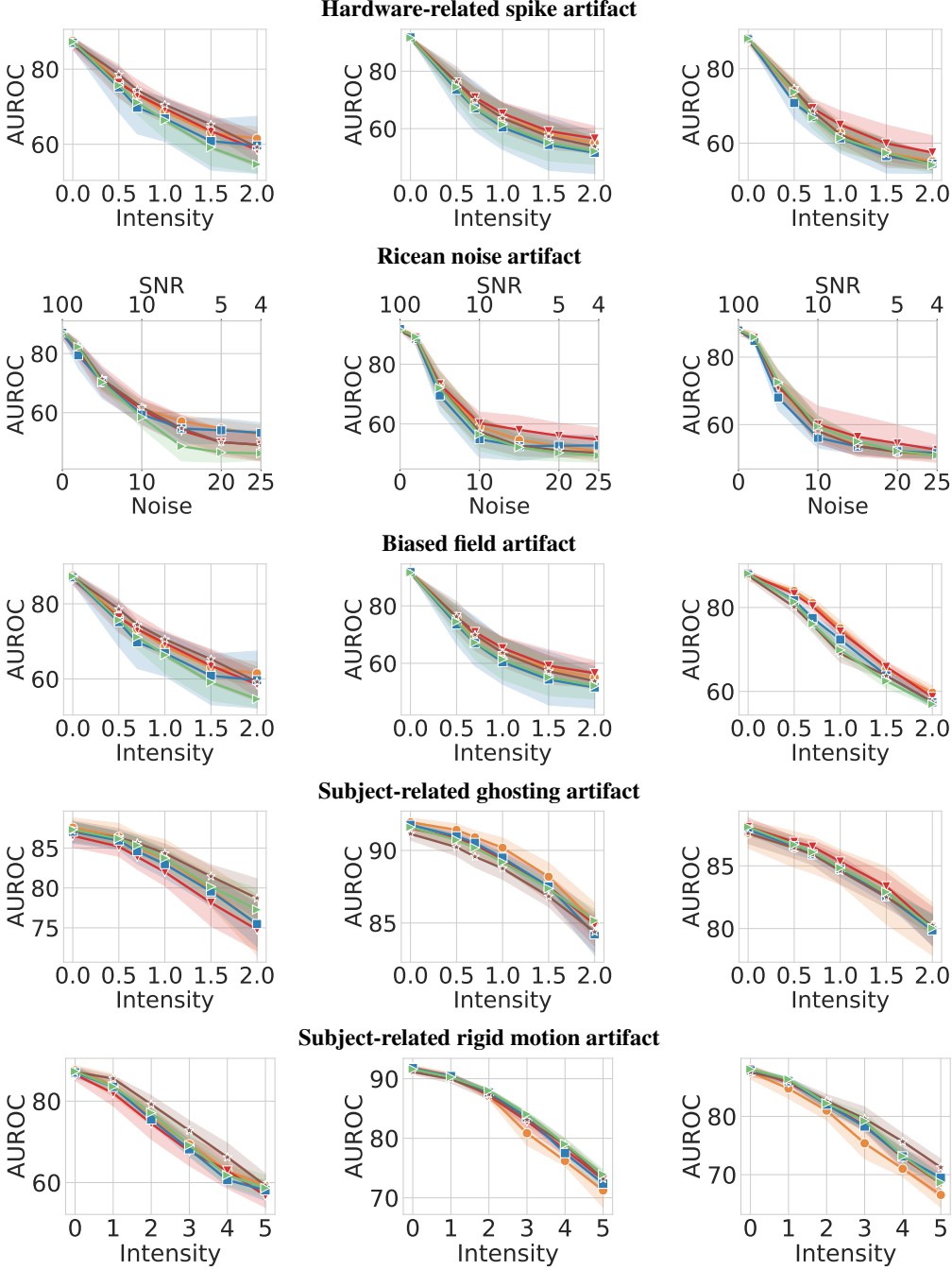

Figure C.14: Comparison of PreactResNet-18 trained with batch size 8, 16, 32, 64 and 128 on ACL (**left**), Meniscus Tear (**middle**), and cartilage (**right**).

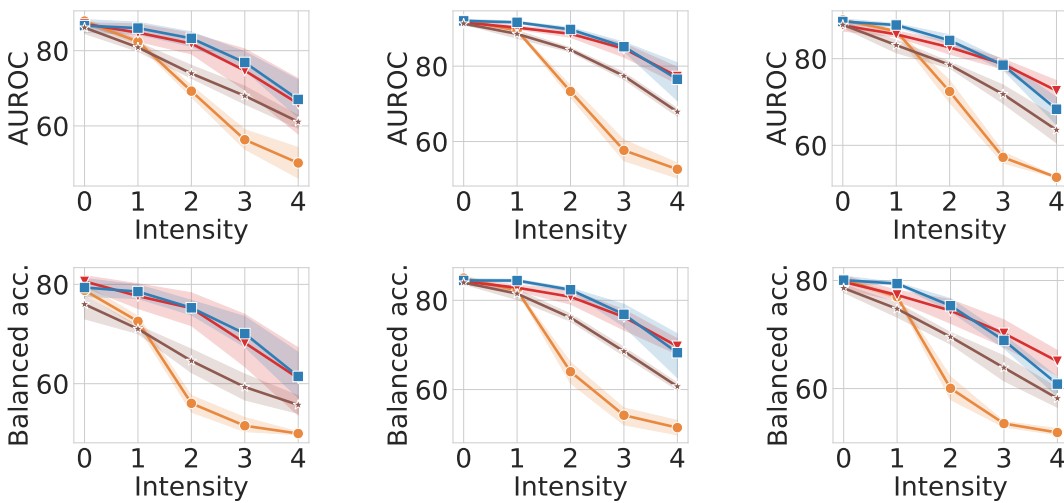

Figure C.15: Comparison of PreactResNet-18 trained with **batch normalization**, **group normalization**, **layer normalization**, and **adaptive batch normalization** on ACL (**left**), Meniscus Tear (**middle**), and cartilage (**right**) on the combination of ricean and ghosting artifact.

