# OpenReview forum: "On Sensitivity and Robustness of Normalization Schemes to Input Distribution Shifts in Automatic MR Image Diagnosis"
_MIDL.io/2023/Conference — MIDL 2023 Poster_

### Official Review · Reviewer_x1or · 2023-02-01

**Confidence:** 5
**Preliminary Rating:** 3
**Recommendation:** Poster

**Summary:**

The authors evaluate the robustness of trained DNN models with regards to out-of-distribution testing samples comparing models using batch normalization (BN) layers to models using alternatives such as group and layer normalization. The performance of the normalization methods are further explored by comparing them to the mean and variance shifts introduced during testing.

**Strengths:**

Concerns surrounding the use of batch normalizations are many, without a clear explanation on when it works and when it doesn't. The explored problem is therefore important and the performed experiments hold merit. The results clearly show the benefits of the proposed alternatives, and raise awareness to the concerns around batch normalization.

**Weaknesses:**

The concerns around batch normalization, frequently discussed in related literature, are not properly addressed. Additionally, the performed evaluations give an unfair disadvantage to batch normalization, by not describing the effects of the batch sized used during evaluation.

**Deanonymize Review:**

no

**Detailed Comments:**

Major comments:
 - Batch normalization, as suggested by the name, normalizes over the batch, therefore it depends heavily on the batch size. This is mentioned when describing the feature normalization methods, however it is not discussed further. The final results depend heavily on the batch size used, and although the training batch size of 32 is mentioned, this is only in the appendix, but it is a crucial part of the work. It is also unclear if the same batch size was used for evaluation or not. I would suggest moving all discussions around the used batch sizes in the main body of the paper. I would also strongly suggest evaluating the impact of the batch size, as I imagine it highly affects the performance of BN.
 - Bringing awareness to the pitfalls of using BN is certainly important, and the experimental results show a better understanding on how the normalization methods behave, however concerns around BN are very well known and documented, which is not referenced by the authors. Such works include Summers et al. 2019 "Four things everyone should know to improve batch normalization." which performs evaluations with similar insights as in the presented paper, and Singh et al. 2019. Other papers often cite issues with BN such as Henaff et al. 2020, or Rivoir et al. 2022. Therefore concerns are often raised against using BN in CNNs due to the different characteristics between the training and testing data statistics, and high dependence on the used batch sizes, which should be acknowledged by the authors when discussing BN. Due to the mentioned references I would strongly disagree with the statement that "until now, the research community in this field has focused on training DNNs primarily using batch normalization for automated diagnosis". I would suggest that the authors include a discussion on often raised concerns against using BN.
 - The experimental results clearly show the drawbacks of BN for out-of-distribution testing samples, and these samples are also shown to have different feature-wise mean and variance compared to the training data. However these empirical results don't show a fundamental insight into the behavior of BN, therefore I feel like the work has not properly "demystified the reasons for the susceptibility of BN" as stated in the key contributions. I would suggest rephrasing it as "providing insight" or "further explores".
 - The most popular alternative to BN to my knowledge is "Instance Normalization" (Ulyanov et al. 2017) which is not mentioned in the paper. I suggest that the authors also include this method in the evaluations, or describe why the method was not included.
 - It would be very interesting to see the results of a model without any normalization layers.
 - Motion artifacts are common in MRI and they are an important issue for DL-based methods in image analysis, but their augmentation is slightly over-simplified and unrealistic. Motion during scan affects the collected k-space which then leads to a blurry image. Although introducing rotation and translation in the image space is also often used, the surrounding padded values (visible in Figure B.8) should be avoided. The sharp edges between the acquired image slice and the padded background value is not only unrealistic, but also largely affects the image distribution, decreasing its mean and increasing its variance. I would suggest starting this augmentation first by selecting only a part of the image ("zooming in") and then applying random translations and rotations, so that image paddings can be avoided. I think this could be a reason why evaluating on the motion artefacts show different conclusions than the others.

Minor notes:
 - I believe the template used for the paper is slightly different than the MIDL official template. I'm sure this is not an issue, but should be corrected to see if the paper still fits the 8 page limit.
 - 2. Related work: Application of deep learning to MRI. - "problems within the MR pipeline" should be "problems within the MRI pipeline". The same comment also holds for the abstract, the subject "pipeline" is for magnetic resonance imaging, not for magnetic resonance in general.
 - 3. 3.1. Motivation - "magnetic Resonance (MR)" the short notation has been used before, multiple times.
 - 3. 3.1. Motivation - "that combines multiple signals" I'm unsure of what this means, and it should be clarified, do the authors mean that it combines multiple quantitative values such as PD, T1- and T2-relaxation times? Or perhaps that multiple individual signals of different contrasts are often used?
 - 3.3. Simulation of Various Artifacts:  Rician noise artifact. - "arises when transitioning from high-field to liw-field MRI scanners". I'm unsure of what this means and should be clarified. This sounds like there are multiple scanners used, whereas I believe it's only a change in the field strengths of the same MRI scanner.
 - 3.3. Simulation of Various Artifacts: Insensity non-uniformity - "and is difficult to detect by visual inspection" - Looking at Figure B.6, I would say it's easy to detect by visual inspection.
 - Figure 1 is never referenced.

**Paper Type:**

methodological development

**Questions To Address In The Rebuttal:**

Please address the detailed comments, most importantly:

- Why does the related works section not discuss papers exploring similar pitfalls of using Batch Normalization?
- What batch size did the authors use for the evaluations? Please address this (and its importance) in the main body of the paper.

---

### Official Review · Reviewer_QfcX · 2023-02-02

**Confidence:** 4
**Preliminary Rating:** 2

**Summary:**

In order to improve the deep learning generalization performance, different normalization layers were used on medical images under different artifacts. The main contribution of the paper is to present how different normalization layers can influence the performance of the model.

The authors hypothesized that a way to improve the generalization of the model is to use a different normalization layer. In the experiments, knee images with different artifacts were trained under different configurations of normalization layers in the PreactResNet-18 architecture. Their findings showed that group and layer normalization can improve the performance of the network.


**Strengths:**

The authors tried to cover a good variety of MRI artifacts and performed an exhaustive analysis of how the artifacts and normalization layers can impact the performance of the model. This can help to provide a better clue of how to design a model architecture for similar applications.

**Weaknesses:**

The batch size can also interfere with the batch normalization layers, as the authors mentioned. However, all the experiments were performed with a batch size fixed at 32. Although the authors claimed that group and layer normalization presented a better result, in my opinion, a new experiment should be considered before drawing this conclusion: an experiment varying the batch size. Also, a better description of the architecture showing how the normalization layers were placed on the model can help to understand the methodology. This could be added to supplementary material.
One of the main motivations for the creation of batch normalization was because of the “internal covariate shift”. Later another paper [1], claimed that the reason that batch normalization works is because it makes the optimization landscape significantly smoother. In other words, it is still not clear why the normalization layers work. Showing how the evolution of the input distribution through the layers would be very interesting to understand more about how the normalization works, instead of only focusing on the final result. Works like [2,3] already presented how different normalization layers can impact the performance of the model, and in [2] the architecture was even submitted on different datasets achieving great results.

[1] How Does Batch Normalization Help Optimization?
[2] nnU-Net: a self-configuring method for deep learning-based biomedical image segmentation
[3] Normalization in Training U-Net for 2-D Biomedical Semantic Segmentation


**Deanonymize Review:**

no

**Detailed Comments:**

The subtitles of the figures could be better described and the training loss could be added to the supplementary material.

**Paper Type:**

validation/application paper

**Questions To Address In The Rebuttal:**

What is happening with internal covariate shifts along the training? What is the effect of the batch size when the batch normalization is used? What is the convergence time of the model using different normalization layers?

---

### Official Review · Reviewer_UiBf · 2023-02-09

**Confidence:** 3
**Preliminary Rating:** 4
**Recommendation:** Poster

**Summary:**

The author comprehensively discussed the sensitivity of normalization methods on various types distribution shifts of MR images. The material is simulated by adding artifacts (e.g. noise, bias field) via post-processing. The the author compared the classification results by using batch normalization, layer normalization, group normalization and adaptive batch normalization. In conclusion, group normalization is more robust with regard to the several domain shift conditions.

**Strengths:**

- The discussion on the robustness normalization methods is interesting and may have very broad impact on task like domain generalization and domain adaptation.

- The experiment clearly show the advantage of group normalization in terms of robustness.

- The explaination is very clear.

**Weaknesses:**

- Lack of real data test. The experiments in the paper are based upon synthetic artifects simulated by post-processing while the real data distribution shift can be more complex. Hence, it is necessary to include experiment conducted on two different real datasets.

- A brief test on the combination of different artifacts can make the experiment more complete.

- The paper aims to discuss one specific downstream task (diagnosis/classification). To show that the conclusion is solid, experiment on other tasks like segmentation will help.

**Deanonymize Review:**

no

**Paper Type:**

validation/application paper

**Questions To Address In The Rebuttal:**

- Did you try to find the optimal number of groups for group normalization? If no, what is the setting for your experiment?

- I think the batch size can also affect the performance for different normalization methods. For example, BN is said to work better for larger batch size (>32). What is the setting in your experiments?

- Is there an intuition that shows why GN outperforms the BN under domain shift?

- Minor suggestion: In **Further analysis**, I think it is better to use different line type for Figure.3 instead of uning the same color code with normalization methods

---

### Meta-Review · Area_Chair_pMkj · 2023-02-24

**Recommendation:** Accept (Poster)
**Confidence:** 3

**Metareview:**

This validation/application paper studies the effectiveness of various normalization strategies in deep learning such as batch normalization, group normalization and layer normalization in dealing data that is corrupted with artifacts (bias field, motion, hardware related spike, etc). The evaluation in the paper is comprehensive for a conference paper and is a strength. The comparison is performed in the context of a downstream classification task. One of the criticisms raised by a reviewer was that the artifacts are simulated. However, the generated artifacts look realistic and the reviewer who raised the concern appears to be satisfied with the authors' response. Another reviewer noted in their reply to the author response that they are changing their rating to Weak Accept (it was originally Weak reject); however, the actual rating was not updated and remains a weak reject.

Overall, I believe that the study targets an important part of the deep learning architectures for medical image analysis. While the results might be limited to one dataset and might not generalize universally, it highlights this important issue researchers should consider when designing their processing pipelines.